# Antibiotic-Induced Dysbiosis of Microbiota Promotes Chicken Lipogenesis by Altering Metabolomics in the Cecum

**DOI:** 10.3390/metabo11080487

**Published:** 2021-07-28

**Authors:** Tao Zhang, Hao Ding, Lan Chen, Yueyue Lin, Yongshuang Gong, Zhiming Pan, Genxi Zhang, Kaizhou Xie, Guojun Dai, Jinyu Wang

**Affiliations:** 1College of Animal Science and Technology, Yangzhou University, Yangzhou 225009, China; zhangt@yzu.edu.cn (T.Z.); 15138343214@163.com (H.D.); lyy3078539326@163.com (Y.L.); gxzhang@yzu.edu.cn (G.Z.); daigj@yzu.edu.cn (G.D.); jywang@yzu.edu.cn (J.W.); 2Joint International Research Laboratory of Agriculture and Agri-Product Safety, Ministry of Education, Yangzhou University, Yangzhou 225009, China; chenlan9326@163.com; 3College of Veterinary Medicine, Yangzhou University, Yangzhou 225009, China; 4Jiangsu Heying Poultry Breeding Technology Co., Ltd., Yancheng 224300, China; gongyongs1023@163.com; 5Jiangsu Key Laboratory of Zoonosis, Key Laboratory of Prevention and Control of Biological Hazard Factors (Animal Origin) for Agrifood Safety and Quality, Ministry of Agriculture and Rural Affairs of the People’s Republic of China, Yangzhou University, Yangzhou 225009, China

**Keywords:** chicken, lipogenesis, gut microbiota, metabolomics, microbiome

## Abstract

Elucidation of the mechanism of lipogenesis and fat deposition is essential for controlling excessive fat deposition in chicken. Studies have shown that gut microbiota plays an important role in regulating host lipogenesis and lipid metabolism. However, the function of gut microbiota in the lipogenesis of chicken and their relevant mechanisms are poorly understood. In the present study, the gut microbiota of chicken was depleted by oral antibiotics. Changes in cecal microbiota and metabolomics were detected by 16S rRNA sequencing and ultra-high performance liquid chromatography coupled with MS/MS (UHPLC–MS/MS) analysis. The correlation between antibiotic-induced dysbiosis of gut microbiota and metabolites and lipogenesis were analysed. We found that oral antibiotics significantly promoted the lipogenesis of chicken. 16S rRNA sequencing indicated that oral antibiotics significantly reduced the diversity and richness and caused dysbiosis of gut microbiota. Specifically, the abundance of Proteobacteria was increased considerably while the abundances of Bacteroidetes and Firmicutes were significantly decreased. At the genus level, the abundances of genera *Escherichia-Shigella* and *Klebsiella* were significantly increased while the abundances of 12 genera were significantly decreased, including *Bacteroides*. UHPLC-MS/MS analysis showed that antibiotic-induced dysbiosis of gut microbiota significantly altered cecal metabolomics and caused declines in abundance of 799 metabolites and increases in abundance of 945 metabolites. Microbiota-metabolite network revealed significant correlations between 4 differential phyla and 244 differential metabolites as well as 15 differential genera and 304 differential metabolites. Three metabolites of l-glutamic acid, pantothenate acid and *N*-acetyl-l-aspartic acid were identified as potential metabolites that link gut microbiota and lipogenesis in chicken. In conclusion, our results showed that antibiotic-induced dysbiosis of gut microbiota promotes lipogenesis of chicken by altering relevant metabolomics. The efforts in this study laid a basis for further study of the mechanisms that gut microbiota regulates lipogenesis and fat deposition of chicken.

## 1. Introduction

Long-term intensive genetic selection has improved growth rate and meat yield in modern broilers. However, the high growth rate is accompanied by excessive fat deposition. Therefore, excessive fat accumulation (especially abdominal fat) in broilers has been a significant problem for broiler breeders worldwide [1]. Studies have shown that depositing unit fat consumes three times more energy than depositing the same amount of lean meat, and abdominal fat is usually discarded as waste [2]. In addition, excessive deposition of abdominal fat causes huge impacts on the production and consumption of broilers, such as reducing feed conversion efficiency and carcass meat yield, affecting the egg production rate, fertilisation rate and hatching rate, and inducing fatty liver syndrome [3]. Therefore, controlling the excessive deposition of fat and improving the feed conversion efficiency and carcass quality of broilers has become one of the major problems that need to be solved urgently in broiler production [4].

The gut microbiota is a complex microbial community comprised of >100 trillion microbial cells. The host phenotype is regulated by its genes and affected by the “host second genome”—gut microbiota. Gut microbiota and its metabolites influence immune diseases, nutritional metabolism and body behaviour through the gut–liver and gut–brain axis. To explore the relationship between gut microbiota and host phenotype, scholars have proposed the concept of “microbiability”. microbiability is defined as the proportion of microbial variance accounted for phenotypic variance, reflecting the effect of gut microbiota on host phenotype [5,6]. The proposal of “microbiability” provides a new idea to investigate the relationship between gut microbiota, host genome and host phenotype. Wen et al. estimated the microbiability of abdominal fat weight and abdominal fat rate of yellow feathered broilers. The duodenal and caecal microbiota made greater contributions to fat deposition and could separately account for 24% and 21% of the variance in the abdominal fat mass after correcting for host genetic effects [7]. The above results indicate that gut microbiota might play an essential role in regulating chicken lipogenesis and fat deposition. However, the mechanisms by which the gut microbiota regulates lipogenesis and fat deposition of chicken remain largely unknown.

Antibiotic-induced dysbiosis of the gut microbiota has been extremely useful in inferring the causal role of microbial dysbiosis with many host phenotypes [8,9]. In the present study, to explore the effect of gut microbiota on lipogenesis, we depleted the gut microbiota with a mixture of antibiotics. The lipogenesis and gut microbiota structure in antibiotic-treated and control chickens were monitored. A non-targeted metabolomics approach was used to characterise metabolic changes caused by gut microbiota dysbiosis and investigate the possible mechanisms underlying the relationship between gut microbiota and chicken lipogenesis. This study can help improve the understanding of the effect of gut microbiota on the lipogenesis of chicken and provide a theoretical basis for further revealing the molecular mechanism of gut microbiota and its metabolites regulating chicken lipogenesis.

## 2. Results

### 2.1. Antibiotic Treatment Promotes Lipogenesis of Chicken

After three weeks of antibiotic treatment, the body weight (BW), abdominal fat weight (AFW), and abdominal fat rate (AFR) of the experimental group (Normal fat diet with antibiotics, NFDA) were statistically highly significantly increased compared to the control group (Normal fat diet, NFD) (*p* < 0.01). The serum triglyceride (STG), high-density lipoprotein (SHDL), low-density lipoprotein (SLDL) and liver triglyceride (TG) levels of the experimental group were statistically significantly increased compared to the control group (*p* < 0.05). Differences in serum total cholesterol (STC) content were not significant (*p* > 0.05) (Figure 1A). Our results indicated that antibiotic treatment significantly promotes the lipogenesis of chicken.

### 2.2. Antibiotic Treatment Alters the Gut Microbiota

A total of 959 OTUs were identified in 10 samples (Appendix A). We investigated the impact of antibiotic mix on cecal microbiota using 16S rRNA sequencing. We found that the diversity of the microbial community significantly decreased in the antibiotic-treated group, as indicated by observed_species (Sob), abundance-based coverage estimators (ACE), Chao1, Shannon and Simpson indices (Figure 1B). It suggested that antibiotic treatment decreased both the richness and evenness of chicken cecal microbiota. The decreased richness and evenness were also shown by the Rank abundance curves (Figure 1C). The beta diversity analysis demonstrated a notable separation of the antibiotic-treated group from the control (Figure 1D). The differences in distance index within the group between the negative control group and the antibiotic-treated groups were highly significant (*p* < 0.01) (Figure 1E). Then, we analysed the composition of cecal microbiota at phylum and genus levels. Firmicutes (45.4%) and Bacteroidetes (43.7%) were the predominant bacteria in the control group at the phylum level. After antibiotic treatment, the abundances of Firmicutes and Bacteroidetes were significantly decreased, while the abundance of Proteobacteria was highly significantly increased (*p* < 0.01). Proteobacteria (90.4%) became the predominant phylum in the antibiotic-treated group (Figure 2A). At the genus level, *Bacteroides* was the predominant genus in the control group, with an abundance of 20.4%. After antibiotic treatment, the majority of genera in the cecum were effective depleted. *Escherichia-Shigella* and *Klebsiella* genera were significantly increased in abundance and became the predominant genera in the antibiotic-treated group (Figure 2B). Taken together, oral antibiotics significantly altered chicken cecal microbiota.

### 2.3. Antibiotic Treatment Alters Cecal Metabolomics

To investigate the effect of oral antibiotics on cecal metabolomics, we detected the cecal metabolites of chickens in control and antibiotic treatment groups by performing a UHPLC–MS/MS analysis. A total of 19,037 metabolic peaks were detected, including 9907 positive ion peaks and 9130 negative ion peaks. Principal components analysis (PCA) plot and cluster heatmap based on all peaks indicated that NFD-1 was an outlier sample and was excluded from subsequent analysis (Figure 2C,D). Furthermore, the PCA plot and cluster heatmap showed that antibiotic treatment significantly altered the metabolomics profiling of chicken cecum. Finally, a total of 2378 metabolites were identified qualitatively based on the primary and second mass data (Figure 2E, Appendix A).

### 2.4. Identification of Differential Taxa

The Welch’s *t*-test was used to detect taxa (phyla and genera) with differential abundances between control and antibiotic treatment groups. Four differential phyla were identified with false discovery rate (FDR) < 0.05, including Firmicutes, Bacteroidetes, Actinobacteria and Proteobacteria (Figure 3A). The abundance of Proteobacteria was significantly increased while the abundances of Firmicutes, Bacteroidetes and Actinobacteria were significantly decreased in the antibiotic treatment group. The ratio of Firmicutes/Bacteroidetes increased from 0.93 to 22.17. At the genus level, 15 differential genera were identified (FDR < 0.05) (Figure 3B). The abundances of 12 genera were significantly decreased, such as *Bacteroides* and *Alistipes*. Only three genera were significantly increased in abundance, including *Escherichia-Shigella*, *Klebsiella*, and *Proteus*.

### 2.5. Prediction of the Function of Cecal Microbiota

Tax4Fun was used to predict the function of the cecal microbiota of chickens in control and antibiotic treatment groups. The results demonstrated that carbohydrate metabolism and lipid metabolism-related microbiota were enriched, and amino acid metabolism and nucleotide metabolism-related microbiota were depleted in response to antibiotic treatment (Figure 4).

### 2.6. Identification and Functional Annotation of Differential Metabolites

A total of 1744 differential metabolites were identified between control and antibiotic treatment groups by variable influence on projection (VIP) > 1 and *p* < 0.05 (Appendix A). In total, 799 metabolites were significantly enriched, while 945 metabolites were depleted significantly by antibiotic treatment (Figure 5A,B). Kyoto Encyclopedia of Genes and Genomes (KEGG) metabolic pathway analysis significantly annotated 1744 differential metabolites to 23 metabolic pathways, among which 15 were metabolism-related pathways such as amino acid metabolism and lipid metabolism (Figure 5C).

### 2.7. Cecal Microbiota Correlated to Lipogenesis of Chicken

To investigate the correlation between cecal microbiota alteration and lipogenesis of chicken, we analyzed the correlation between species abundance matrix (phylum and genus level) and lipogenesis related indicators by Mantel test. At the phylum level, cecal microbiota was highly significantly correlated with AFW (r = 0.91) and BW (r = 0.74) (*p* < 0.01), and was significantly correlated with AFR (r = 0.81), STG (r = 0.55), SHDL (r = 0.31), SLDL (r = 0.30), and LTG levels (r = 0.66) (*p* < 0.05). At the genus level, cecal microbiota was highly significantly correlated with AFW (r = 0.93), AFR (r = 0.85), BW (r = 0.77), and LTG (r = 0.69) levels (*p* < 0.01), and was significantly correlated with STG (r = 0.52), SHDL (r = 0.29), and SLDL (r = 0.34) levels (*p* < 0.05). No significant correlation was detected between cecal microbiota and STC level (*p* > 0.05) (Figure 6A).

We calculated the Pearson correlation coefficients between differential taxa (phyla and genera) and lipogenesis related indicators to further explore the relationship between cecal microbiota and lipogenesis. The abundance of Proteobacteria was significantly positively correlated with all lipogenesis related indicators except STC level (*p* < 0.01 or *p* < 0.05). The abundances of Firmicutes, Bacteroidetes and Actinobacteria were significantly negatively correlated with most of the lipogenesis indicators (*p* < 0.01 or *p* < 0.05) (Figure 6B). At the genus level, the abundance of *Escherichia-Shigella* was significantly positively correlated with all the eight lipogenesis related indicators, the abundance of *Klebsiella* was significantly positively correlated with seven lipogenesis related indicators except STC level, and the abundance of *Proteus* was significantly positively correlated with BW, AFW, AFR, STG, and STC levels (*p* < 0.05 or *p* < 0.01). Conversely, the abundances of 12 genera, including *Bacteroides* and *Ruminoccoccus*_*torques*_*group* were significantly negatively correlated with most of the lipogenesis related indicators (*p* < 0.05 or *p* < 0.01) (Figure 6C).

### 2.8. Integrated Microbiome and Metabolomics Analysis

To explore whether the altered gut microbiota regulates lipogenesis of chicken by affecting cecal metabolomics, we performed an integrated microbiome and metabolomics analysis based on the differential taxa and metabolites. At the phylum level, four differential phyla were significantly correlated with 244 differential metabolites (|r| > 0.8, *p* < 0.05). Specifically, the Proteobacteria phylum was positively correlated with 103 differential metabolites and negatively correlated with ten differential metabolites. The Bacteroidetes phylum was positively correlated with 93 differential metabolites and negatively correlated with five differential metabolites. The Firmicutes was positively correlated with 110 differential metabolites and negatively correlated with six differential metabolites. The Actinobacteria phylum was positively correlated with 103 differential metabolites and negatively correlated with 11 differential metabolites (Figure 7, Appendix A).

At the genus level, 15 differential genera were significantly correlated with 304 differential metabolites (|r| > 0.8, *p* < 0.05). Three genera increased in abundance after antibiotic treatment, including *Escherichia-Shigella*, *Klebsiella* and *Proteus*, were positively correlated with 194 differential metabolites (r > 0.8, *p* < 0.05) and negatively correlated with 20 differential metabolites (r < −0.8, *p* < 0.05) (Figure 8, Appendix A). Twelve genera were depleted by antibiotic treatment, and were positively correlated with 238 differential metabolites (r > 0.8, *p* < 0.05) and negatively correlated with 60 differential metabolites (r < −0.8, *p* < 0.05) (Figure 9, Appendix A).

### 2.9. Metabolite Set Enrichment Analysis of Metabolites in the Networks

To identify metabolites that link cecal microbiota with the lipogenesis of chicken, we performed metabolite set enrichment analysis on the metabolites involved in the networks using MetaboAnalyst 5.0 based 44 metabolite sets in faeces. At the phylum level, three metabolite sets were significantly enriched (*p* < 0.05), including obesity, myalgic encephalomyelitis/chronic fatigue syndrome and unclassified Ibd (Figure 10A). Four metabolite sets were significantly enriched (*p* < 0.05) at the genus level, including unclassified Ibd, myalgic encephalomyelitis/chronic fatigue syndrome, gout and obesity (Figure 10B). We found that the obesity metabolite set was significantly enriched at both the phylum and genus levels. Three differential metabolites were involved in the obesity set: l-glutamic acid, pantothenic acid, and *N*-acetyl-l-aspartic acid. The abundances of l-glutamic acid and pantothenic acid, which were significantly increased following depletion of cecal microbiota by oral antibiotics (Figure 10E), were highly significantly positively correlated with the abundances of phylum Proteobacteria and genus *Escherichia-Shigella* (*p* < 0.01) and significantly negatively correlated with the abundance of genus *Klebsiella* (*p* < 0.05) (Figure 10C,D). A highly significant positive correlation was observed between the abundances of l-glutamic acid and genus *Proteus* (*p* < 0.01) (Figure 10D). *N*-acetyl-l-aspartic acid was significantly depleted in abundance by antibiotic treatment (*p* < 0.01). The abundance of *N*-acetyl-l-aspartic acid was highly significantly positively correlated with the abundances of phylum Bacteroidetes (*p* < 0.01) and significantly positively correlated with the abundances of phyla Firmicutes and Actinobacteria (*p* < 0.05). At the genus level, the abundance of *N*-acetyl-l-aspartic acid was highly significantly positively correlated with the abundances of *Phascolarctobacterium* and *Lachnospiraceae* (*p* < 0.01) and significantly positively correlated with the abundances of *Bacteroides*, *Ruminococcus*_*torques*_*group*, *Desulfovibrio*, *Shuttleworthia*, *Olsenella*, *Megasphaera*, *Butyricicoccus* and *Blautia* (*p* < 0.05).

## 3. Discussion

Clarifying the mechanisms of lipogenesis is essential for the improvement of the fat trait in chicken. Studies have revealed that gut microbiota participates in the regulation of lipogenesis and fat deposition in chicken [7,10]. However, the precise molecular mechanisms by which gut microbiota regulates lipogenesis and fat deposition remain largely unknown. In the present study, we depleted the gut microbiota of chicken by oral antibiotic mix. The changes in lipogenesis indicators, cecal microbiota and metabolomics were detected. The correlation between cecal microbiota and metabolomics and lipogenesis indicators were analysed.

A previous study has shown causality between antibiotic treatment and lipogenesis in chicken [11]. Consistent with these studies, our research also indicated that oral antibiotics could significantly promote lipogenesis and fat deposition in chicken. The antibiotic mix has been widely used in depleting gut microbiota and constructing gut microbiota dysbiosis models [12,13]. Thus, we depleted the gut microbiota of chicken with oral antibiotics. The 16S rRNA sequencing results showed that oral antibiotics resulted in decreased richness and evenness of gut microbiota and induced gut microbiota dysbiosis. It suggests that oral antibiotics could be a useful tool in depleting gut microbiota and constructing the gut microbiota dysbiosis model of chicken.

In this study, the Tax4Fun algorithm was used to predict the potential function of the cecal microbiota of chickens. Bacteria involved in carbohydrate metabolism and lipid metabolism-related microbiota were significantly enriched in antibiotic-treated chickens. In contrast, bacteria involved in amino acid metabolism and nucleotide metabolism-related microbiota were significantly depleted in response to antibiotic treatment. This suggests that the antibiotic-induced dysbiosis of cecal microbiota might promote carbohydrate metabolism and lipid metabolism and inhibit amino acid metabolism and nucleotide metabolism of chicken.

Then, we further investigated whether the antibiotic-induced dysbiosis of gut microbiota was associated with lipogenesis of chicken based on 16S rRNA sequencing data. Recent studies have found that obesity has been associated with reduced abundance of phylum Bacteroidetes, increased Firmicutes/Bacteroides ratio and increased abundance of phylum Proteobacteria [14,15,16]. In accordance with previous studies, we also found that oral antibiotics promote lipogenesis and fat deposition of chicken, accompanied by the significantly increased abundance of Proteobacteria, decreased abundance of Bacteroides, and decreased Firmicutes/Bacteroidetes ratio. Association analysis demonstrated a significant positive correlation between lipogenesis and the abundance of Proteobacteria, and significant negative correlations between lipogenesis and the abundances of Bacteroidetes as well as Firmicutes. The above results indicated that Proteobacteria might promote lipogenesis of chicken, while Bacteroidetes has the opposite effect.

Twelve genera were significantly depleted by antibiotic treatment at the genus level, including *Escherichia-Shigella*, *Klebsiella* and *Proteus*. *Escherichia-Shigella*, *Klebsiella*, *Bacteroides* and *Ruminococcus* are reported to be associated with lipogenesis and lipid metabolism. The microbiota-depleted mice with a single bacterium (*Escherichia-Shigella*) after antibiotic treatment were resistant to Fuzhuan Brick Tea induced antiobesity and metabolic improvement [17]. *Klebsiella* was significantly enriched in the faeces of obese children, and its abundance was positively correlated with serum TG, TC and LDL levels [18]. In the present study, *Escherichia-Shigella*, and *Klebsiella* were significantly enriched in the cecum of antibiotic-treated chickens, and their abundances were positively linked to lipogenesis indicators, suggesting their potential roles in promoting lipogenesis of chicken. *Bacteroides* was reported to be associated with fat deposition in pigs. The abundance of *Bacteroides* decreased significantly with the development and was negatively significantly correlated with lipogenesis and fat deposition of pigs [19]. Hou et al. analysed the faecal metagenomes of the divergently selected lean and fat line chickens and found that *Bacteroides*, *Ruminococcus*, *Butyricicoccus* and *Blautia* have significantly higher abundances in the lean line than in the fat line [20]. *Ruminococcus* has also been shown to mediate the function of branched-chain amino acids in reducing the hepatic fat accumulation of high-fat diet-induced mice [21]. The studies mentioned above showed that genera *Bacteroides*, *Ruminococcus*, *Butyricicoccus*, *Blautia* were related to lipogenesis and fat deposition in animals and humans. However, little is known about their functions in the lipogenesis of chicken. Our study demonstrated that the abundance of *Bacteroides*, *Ruminococcus*, *Butyricicoccus*, *Blautia* were negatively significantly correlated with lipogenesis, suggesting their potential roles in inhibiting excessive lipogenesis and fat deposition in chicken.

This study mainly aimed to explore the mechanisms underly the role of gut microbiota in regulating the lipogenesis of chicken. Studies have shown that gut microbiota could regulate host lipogenesis through its metabolic products. To ask whether the antibiotic-induced dysbiosis of gut microbiota alters gut metabolomics, which affects lipogenesis and fat deposition in chicken, we detected the gut metabolomics of control and antibiotic-treated chickens. We analysed correlations between gut metabolomics and lipogenesis. Four differential phyla were significantly correlated with 244 differential metabolites, and fifteen differential genera were significantly correlated with 304 differential metabolites. Metabolite set enrichment analysis showed that a set of metabolites were significantly enriched into obesity, including l-glutamic acid, Pantothenic acid, and *N*-acetyl-l-aspartic acid. Glutamine is the primary source of intestinal cells. It can maintain the structural integrity of the gut and improve the intestinal digestive and absorptive functions. In addition, l-glutamine acts as a substrate in protein synthesis and participates in the regulation of lipid metabolism [22]. In pigs, dietary supplementation of glutamic acid decreased the backfat thickness. Dietary supplementation with both arginine and glutamine increases the IMF deposition, improves the meat colour and fatty acid composition, and affects fatty acid content, flavour compounds, and sensory quality of pork [23,24]. Pantothenate acid is a member of the vitamin B family. It is a cofactor of Coenzyme A (CoA) and a part of acyl carrier proteins (ACPs). Pantothenate acid was involved in the metabolisms of carbohydrates, fat, and protein, especially the synthesis and metabolism of fats [25]. In goose, dietary supplementation with pantothenate acid can significantly affect lipid [26]. Fat and energy stores were significantly reduced when chicks were fed diets deficient in pantothenic acid [27]. There are currently no published reports of *N*-acetyl-l-aspartic acid associated with lipid metabolism in the animal. In summary, l-glutamine acid and pantothenate acid abundances were significantly decreased, while the abundance of *N*-acetyl-l-aspartic acid was significantly increased following antibiotic treatment. These three metabolites were significantly correlated with the altered cecal microbiota and lipogenesis, suggesting their potential roles in mediating the enhanced lipogenesis regulated by antibiotic-induced dysbiosis of gut microbiota in chicken.

## 4. Materials and Methods

### 4.1. Animal

Twenty 4-week old Heying chickens with similar body weight were randomly divided into two groups, control (*n* = 10) and antibiotic-treated (*n* = 10), respectively. There were no differences in body weight between the two groups (*p* = 0.70).

### 4.2. Antibiotic Treatment

Each chicken was housed in a separate cage with water and food ad libitum. An antibiotic mix consisting of vancomycin (Macklin, Shanghai, China, 0.5 μg/mL), Ampicillin (Macklin, Shanghai, China, 1 μg/mL), Neomycin sulfate (Macklin, Shanghai, China, 1 μg/mL) and Metronidazole (Macklin, Shanghai, China, 1 μg/mL) was used to deplete the gut microbiota. The chickens were treated with a mix of four antibiotics or distilled water by oral gavage for 21 days. The dosing of antibiotics in the drinking water refers to previous studies [12,28]. The body weight of the chicken was measured after three weeks of antibiotic treatment.

### 4.3. Sample Collection

After 21 days of antibiotic treatment, five healthy chickens with similar body weight were selected from the control group (NFD, *n* = 5), and five healthy chickens with similar body weight were selected from the antibiotic-treated groups (NFDA, *n* = 5), respectively. Blood samples were collected by venipuncture, and serum was obtained by centrifugation and stored at 4 °C. The liver tissues were collected, placed in microcentrifuge tubes, flash-frozen in liquid nitrogen and stored at −80  °C until further analysis. The abdominal fat tissues were isolated, and the AFW was weighed to calculate the abdominal fat rate (AFR): AFR = AFW/BW × 100%. The cecal contents were collected by gently squeezing the contents from the tissue into a sterile collection tube.

### 4.4. Triglyceride Content Detection in Liver

About 0.1 g of each liver sample was homogenised using heptane and isopropanol (1:1, *v*/*v*) as the solvent. The homogenate was centrifuged for 10 min at 8000× *g*, 4 °C. The supernatant TG content (mmol/g) was then measured by TG kit (Solarbio, Beijing, China) at the absorbance of 450 nm according to the manufacturer’s instructions.

### 4.5. Measurement of Serum LDL Oxidation, TC, and TG

The STG, STC, SHDL and SLDL levels were measured using TG, TC, HDL-C, and LDL-C assay kit (Jiancheng, Nanjing, China) according to the manufacturer’s instructions.

### 4.6. 16S rRNA Sequencing

The microbial DNA was extracted from cecal contents from ten chickens using HiPure stool DNA kits (Magen, Guangzhou, China) according to the manufacturer’s instructions. The V3 and V4 region of the 16S rDNA gene was amplified with specific barcoded primers. The primer sequences were as follows: 341F: 5′-CCTACGGGNGGCWGCAG-3′, 806R: 5′-GGACTACHVGGGTATCTAAT-3′. The PCR program was 94 °C for 2 min, followed by 30 cycles of 98 °C for 10 s and 66 °C for 30 s, and final extension at 72 °C for 1 min. The amplification products were purified using AMPure XP beads (Beckman Coulter, Brea, CA) and quantified using ABI StepOnePlus RealTime PCR System. Then, 16S rRNA high-throughput sequencing was performed using an Illumina Novaseq 6000 PE250 platform.

The raw data were subjected to quality control using FASTP software (version 0.18.0) [29]. Reads containing more than 10% unknown nucleotides and low quality reads containing more than 50% low quality (Q-value ≤ 20) bases were removed. The paired-end clean reads were merged as raw tags using FLSAH (version 1.2.11) [30] with a minimum overlap of 25 bp and 2% mismatches. High-quality clean tags were obtained by filtering noisy sequences of raw tags using QIIEM software (version 1.9.1) [31]. Then, the clean tags were mapped to the reference database (version 4.2, http://drive5.com/uchime/uchime_download.html, accessed on 5 December 2020). The effective tags were obtained by filtering chimeric tags using the UCHIME algorithm [32]. Effective tags with ≥97% similarity were assigned to the same operational taxonomic units (OTU) using UPARSE (version 9.2.64) [33]. For each resulting OTU, the most abundant tags were selected as a representative sequence.

The representative OTU sequences were classified into organisms by a naive Bayesian model using an RDP classifier (version 2.2) based on the SILVA database [34], with the confidence threshold value of 0.8. Heatmap of species abundance was plotted using the pheatmap package (version 1.0.12, https://CRAN.R-project.org/package=pheatmap, accessed on 5 December 2020) in R project (version 4.0.4, www.rproject.org, accessed on 5 December 2020) Sobs, Shannon, Simpson, Chao1 and ACE indices were calculated using QIIME (version 1.9.1). Alpha index comparison between groups was calculated by Welch’s *t*-test using the R project Vegan package (version 2.5.3, http://CRAN.R-project.org/package=vegan, accessed on 6 December 2020). Sequence alignment was performed using Muscle (version 3.8.31) [35], and a phylogenetic tree was constructed using FastTree (version 2.1) [36]. Then unweighted unifrac distance matrix was generated by the GuniFrac package (version 1.0) [37] in the R project. Multivariate statistical techniques including principal coordinates analysis (PCoA) and non-metric multi-dimensional scaling (NMDS) of unweighted unifracs distances were generated in the R project Vegan package (version 2.5.3) and plotted in R project ggplot2 package (version 2.2.1, https://doi.org/10.1007/978-3-319-24277-4_9, accessed on 6 December 2020). Welch’s *t*-test was used to compare the beta diversity metrics between groups. The Welch’s *t*-test was used to detect taxa with significant differential abundances at the phylum and genus levels. Significant differences were declared at FDR < 0.05. The Tax4Fun algorithm was applied to predict the function of the gut microbiota [38]. The 16S rRNA sequencing data were deposited in the Genome Sequence Archive (GSA) database under accession no CRA004512.

### 4.7. Metabolite Extraction and UHPLC-MS/MS Analysis

The ten chickens’ cecal contents samples (50 mg) were collected in 1.5-mL Eppendorf microcentrifuge tubes, and extract solvent (acetonitrile–methanol–water, 2:2:1, 1000 μL) was then added. The mixture was then vortexed for 30 s, homogenised at 45 Hz for 4 min, sonicated for 5 min in an ice-water bath, and incubated at −20 °C for 1 h prior to centrifugation 12,000 rpm at 4 °C for 15 min. The supernatants (200 μL) were transferred to an injection bottle for UHPLC–MS/MS analysis. UHPLC–MS analysis was performed using a UHPLC system (1290, Agilent Technologies) with a UPLC HSS T3 column (2.1 mm × 100 mm, 1.8 μm) coupled to a Q Exactive benchtop Orbitrap mass spectrometer (Orbitrap MS, Thermo). Samples pooled by mixing an equal aliquot of the supernatants from all samples were used as quality control (QC) samples.

### 4.8. Metabolomics Data Analysis

Raw data obtained from the UHPLC-MS/MS platform were converted to the mzML format using ProteoWizard, and then processed by R package XCMS (version 3.2) [39] with retention time alignment, peak detection, and peak matching. A feature table consisting of the retention time (Rt), mass-to-charge ratio (m/z) values, and peak intensity was obtained that indicated the peak areas for each feature across all samples. Peak annotation for all features was performed using OSI-SMMS (version 1.0) with an in-house MS/MS database [40]. Identification of metabolites must meet two criteria: accurate mass with variation less than 10 ppm and high MS/MS score > 0.6. By analysing OPLS-DA loadings, the differential metabolites were identified by VIP > 1 and *p* < 0.05. The heatmap of metabolites was plotted using the R package pheatmap. Pathway enrichment of the differential metabolites was performed using MetaboAnalyst 5.0 [41]. Pathway with *p* < 0.05 was considered significantly enriched. Raw metabolomics data are available in the Metabolights Database at https://www.ebi.ac.uk/metabolights/MTBLS3089, accessed on 7 July 2020.

### 4.9. Associations between Gut Microbiota and Lipogenesis

Associations between gut microbiota (phylum and genus level) and lipogenesis (AFW, AFR, BW, liver TG level, STG level, STC level, SHDL level, and SLDL level) were analysed by Mantel test in R project Vegan package (version 2.5.3). Pearson correlation analysis of relationships between differential phylum/genera and lipogenesis (AFW, AFR, BW, liver TG level, STG level, STC level, SHDL level, and SLDL level) was analysed using psych package in R software (version 1.8.4, https://personality-project.org/r/psych, accessed on 5 April 2020). Correlation results were reported significant if the *p*-value  <  0.05.

### 4.10. Integrated Microbiome and Metabolomics Analysis

The correlations between differential phylum/genera and differential metabolites were measured using Pearson correlation analysis. Phylum/genera-metabolite pairs with |r| > 0.8 and *p* < 0.05 were considered significant. The phylum-metabolite and genus-metabolite networks were constructed based on the significantly correlated pairs and visualised using Gephi software (version 0.9.2) [42]. The metabolites in the constructed networks were analysed for metabolite set enrichment analysis (MSEA) using MetaboAnalyst 5.0. Metabolite set with *p* < 0.05 was considered significant.

## 5. Conclusions

In summary, our study indicated that oral antibiotics significantly altered gut microbiota and metabolomics in chicken. The antibiotic-induced dysbiosis of gut microbiota promotes the lipogenesis of chicken. Metabolites might mediate this function of gut microbiota in lipogenesis in the cecum, such as l-glutamic acid, Pantothenic acid and *N*-acetyl-l-aspartic acid. Our findings provide a theoretical basis for further understanding the mechanism underlying the role of gut microbiota in regulating the lipogenesis of chicken.

## Figures and Tables

**Figure 1 metabolites-11-00487-f001:**
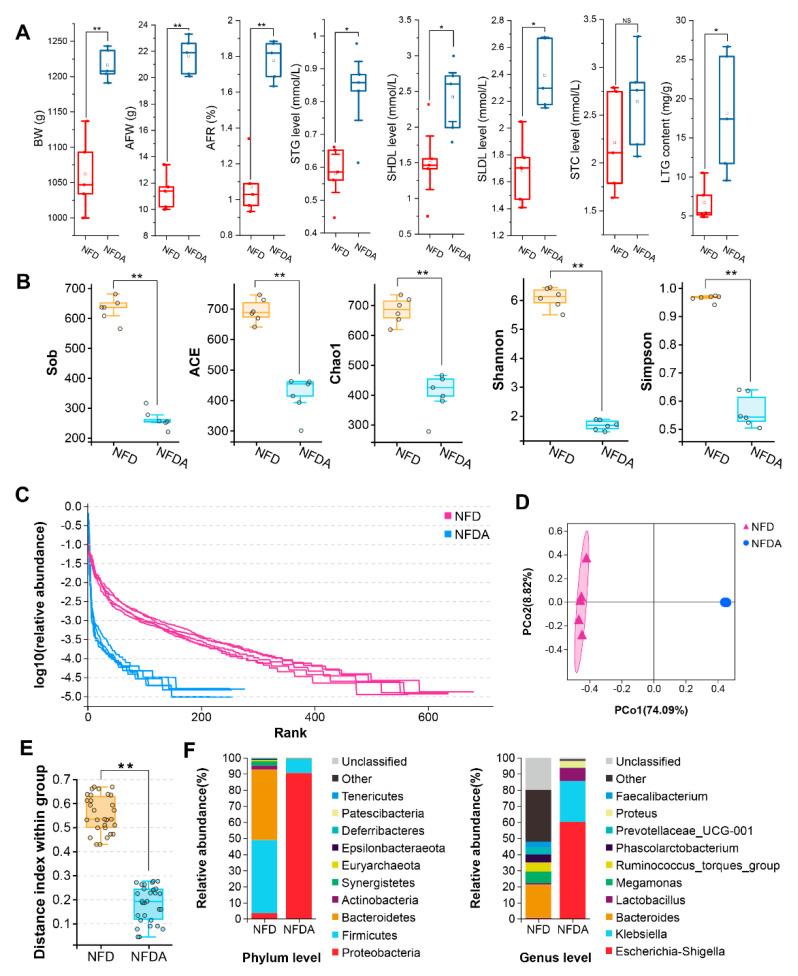
The impact of antibiotic treatment on lipogenesis and cecal microbiota of chicken. (**A**) BW, AFW, AFR, STG level, SHDL level, SLDL level and liver TG level of chickens in antibiotic-treated (*n* = 5) and control (*n* = 5) groups. Oral antibiotics significantly promote the lipogenesis of chicken. (**B**) Comparison of alpha diversity indices of cecal microbiota between control and antibiotic treatment groups. (**C**) Comparison of Rank curve of cecal microbiota between control and antibiotic treatment groups. (**D**) PCoA of beta diversity based on the unweighted UniFrac analysis of the operational taxonomic units (OTU) level. (**E**) Significance test of differences in beta diversity of cecal microbiota between control and antibiotic treatment groups using Welch’s *t*-test. (**F**) Cecal microbiota composition at the phylum and genus levels. * *p* < 0.05, ** *p* < 0.01, NFD (*n* = 5), NFDA (*n* = 5).

**Figure 2 metabolites-11-00487-f002:**
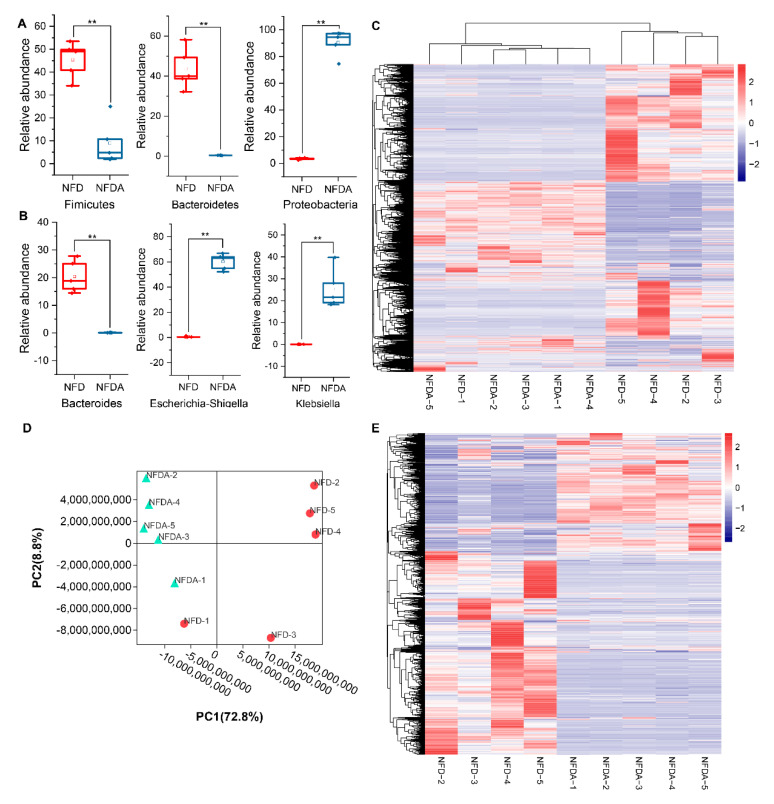
The effect of antibiotic treatment on cecal microbiota and metabolomics. (**A**) The impact of antibiotic treatment on cecal microbiota at the phylum level. The abundances of Firmicutes and Bacteroidetes were highly significantly depleted by antibiotic treatment, while the abundance of Proteobacteria was highly significantly increased (*p* < 0.01). (**B**) The impact of antibiotic treatment on cecal microbiota at the genus level. The antibiotic treatment presented a highly significantly decreased abundance of *Bacteroides* genus and an increased abundance of *Escherichia-Shigella* and *Klebsiella* genera (*p* < 0.01). (**C**) Clustering heatmap of all peaks. The red, white, and blue colours represent the expression level from high to low. (**D**) PCA plot of the samples based on all peaks. NFD (*n* = 5), NFDA (*n* = 5). (**E**) Clustering heatmap of the identified 2378 metabolites across all samples (NFD (*n* = 4), NFDA (*n* = 5)). ** *p* < 0.01.

**Figure 3 metabolites-11-00487-f003:**
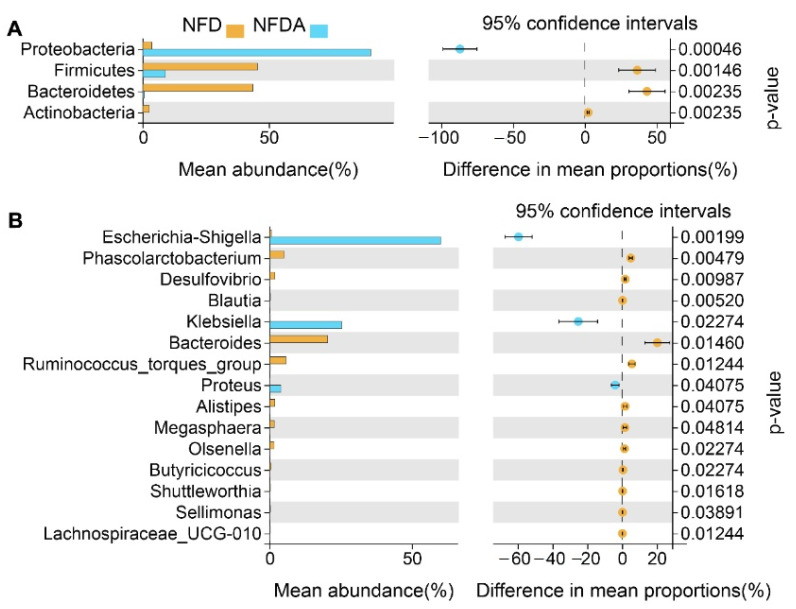
Differential taxa (phyla and genera) between control (*n* = 5) and antibiotic treatment (*n* = 5) groups. (**A**) Differential phyla. (**B**) Differential genera. *p* < 0.05 indicates significant difference. NFD (*n* = 5), NFDA (*n* = 5).

**Figure 4 metabolites-11-00487-f004:**
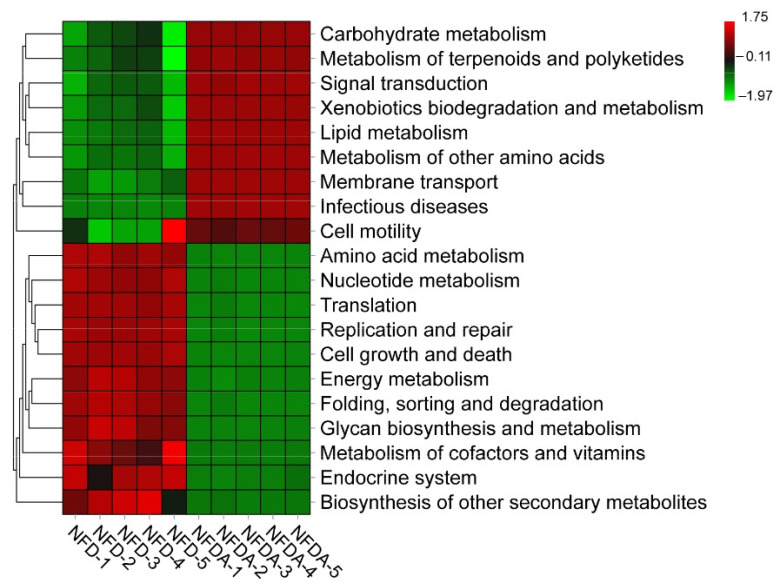
Prediction of the function of cecal microbiota. The red, dark and green colours represent enrichment degrees from high to low. NFD (*n* = 5), NFDA (*n* = 5).

**Figure 5 metabolites-11-00487-f005:**
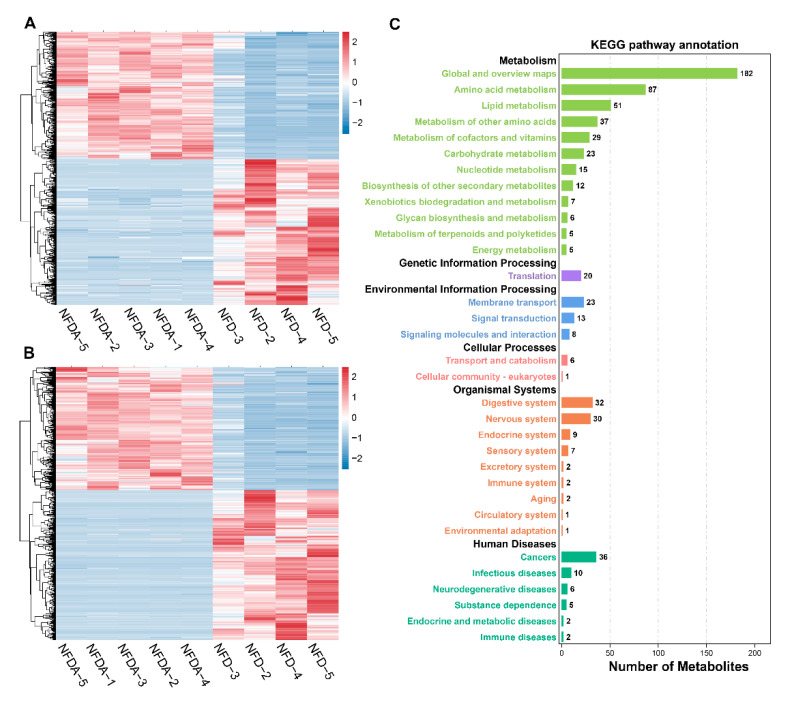
Identification and functional annotation of differential metabolites. (**A**) Clustering heatmap of differential metabolites in positive ion mode. (**B**) Clustering heatmap of differential metabolites in negative ion mode. The red, white and blue colours represent relative abundance from high to low. (**C**) KEGG pathway annotation of differential metabolites. NFD (*n* = 4), NFDA (*n* = 5).

**Figure 6 metabolites-11-00487-f006:**
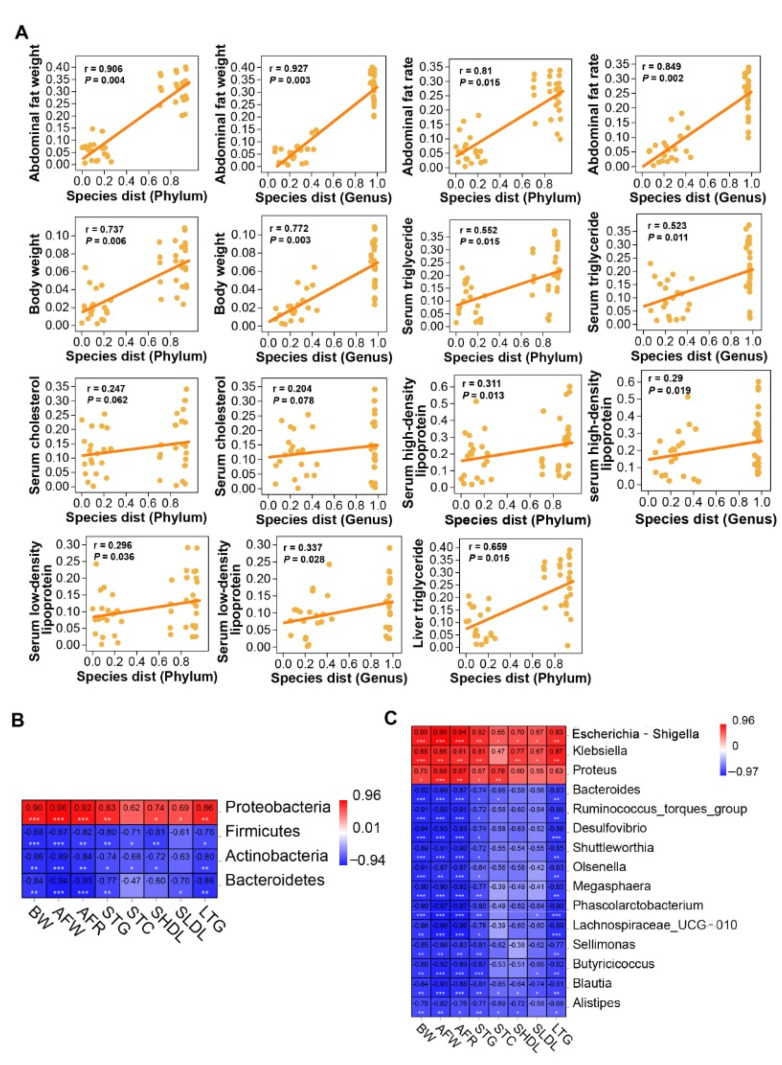
Association between cecal microbiota and lipogenesis. (**A**) Association between species abundance matrix (phylum and genus levels) and lipogenesis indicators. *p* < 0.05 represents significant difference, *p* < 0.01 represents highly significant difference. (**B**) Heatmap of correlation between differential phyla and lipogenesis indicators. (**C**) Heatmap of correlation between differential genera and lipogenesis indicators. * *p* < 0.05, ** *p* < 0.01. *** *p* < 0.001. The red and blue colours indicate positive and negative correlation, respectively. NFD (*n* = 5), NFDA (*n* = 5).

**Figure 7 metabolites-11-00487-f007:**
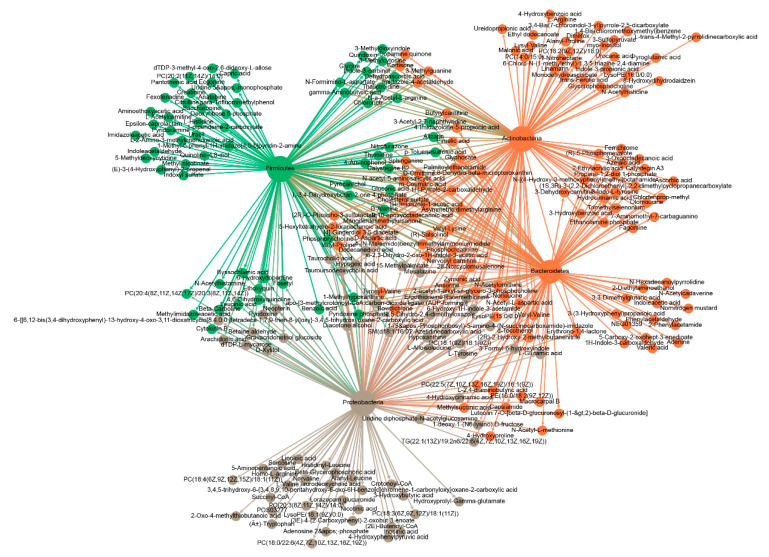
Correlation network of differential phyla and metabolites. Different colours represent different clustering modules. The Spearman correlation coefficients between differential phyla and metabolites were calculated, and pairs with |r| > 0.8 and *p* < 0.05 were used to construct the correlation network. NFD (*n* = 4), NFDA (*n* = 5).

**Figure 8 metabolites-11-00487-f008:**
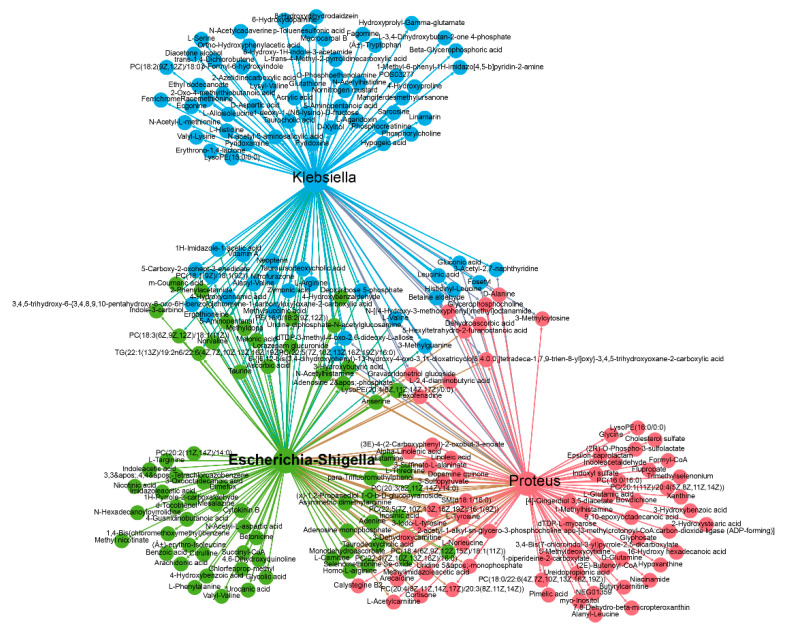
Correlation network between *Escherichia-Shigella*, *Klebsiella* and *Proteus* and differential metabolites. Different colours represent different clustering modules. NFD (*n* = 4), NFDA (*n* = 5).

**Figure 9 metabolites-11-00487-f009:**
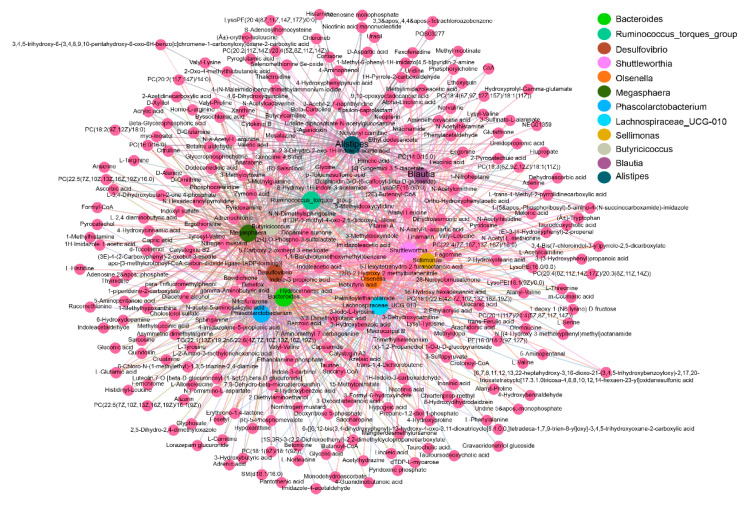
Correlation network between genera depleted by antibiotic treatment and differential metabolites. Different colours represent different genera. The wine-red nodes represent differential metabolites. NFD (*n* = 4), NFDA (*n* = 5).

**Figure 10 metabolites-11-00487-f010:**
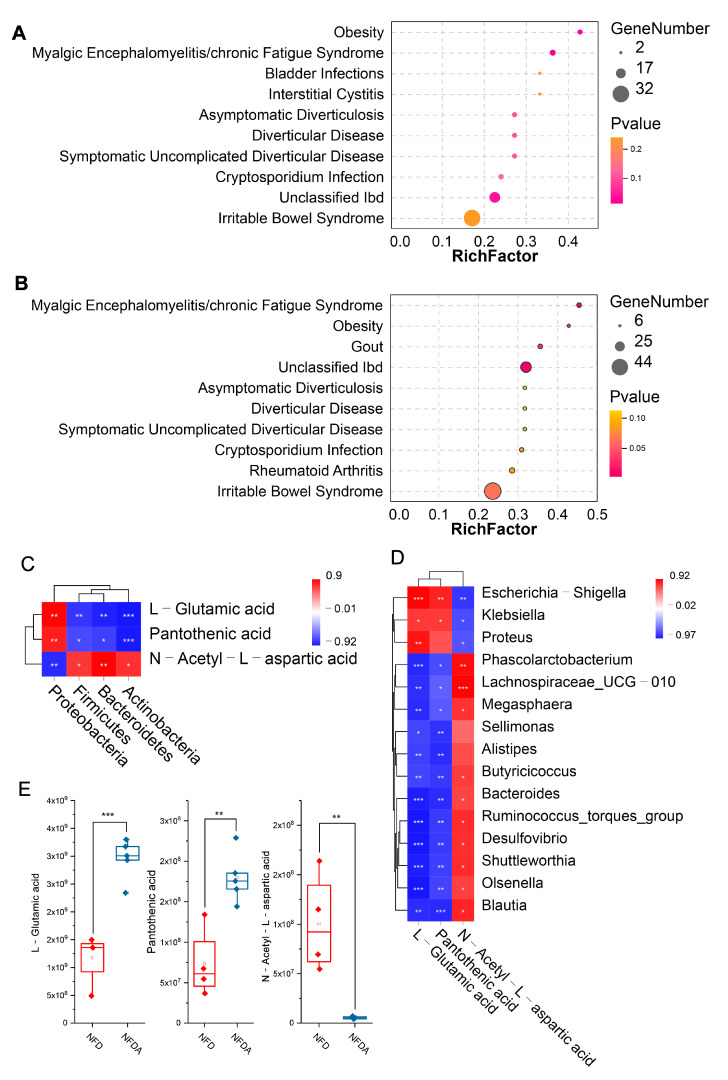
Identification of key metabolites that link cecal microbiota and lipogenesis. (**A**) Metabolite set enrichment analysis of differential phyla-related metabolites. (**B**) Metabolite set enrichment analysis of differential genera-related metabolites. (**C**) Correlation heatmap of obesity-related metabolites and differential phyla. (**D**) Correlation heatmap of obesity-related metabolites and differential genera. (**E**) The abundance of obesity-related metabolites across samples. The node size represents the number of metabolites. The orange and wine-red colours indicate *p*-value from high to low. In the heatmap, blue and red colours represent negative and positive correlations, respectively. * *p* < 0.05, ** *p* < 0.01, *** *p* < 0.001, NFD (*n* = 4), NFDA (*n* = 5).

## Data Availability

Data available in a publicly accessible repository. The 16S rRNA data presented in this study are openly available in [Cecal microbiome of chickens treated by antibiotics] at [the Genome Sequence Archive (GSA) database ], reference number [CRA004512]. The metabolomic data presented in this study are openly available in [Antibiotic-induced dysbiosis of microbiota promotes chicken lipogenesis by altering metabolomics in the cecum] at [https://www.ebi.ac.uk/metabolights/MTBLS3089], reference number [MTBLS3089].

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
