# Peer review of "Antibiotic-Induced Dysbiosis of Microbiota Promotes Chicken Lipogenesis by Altering Metabolomics in the Cecum"

_metabolites, 2021, doi:10.3390/metabo11080487_

Round 1

Reviewer 1 Report

In the manuscript of Zhang et al. entitled “Antibiotic-induced dysbiosis of microbiota promotes chicken 2 lipogenesis by altering metabolomics in the cecum”, the correlation between antibiotic-induced dysbiosis of gut microbiota and metabolites and lipogenesis were analyzed. This study is well organized. The experimental design, procedure for assays are well described, and the results are well documented with good quality data. Some minor concerns are listed below:

  1. Five chickens were randomly selected from the control and antibiotic-treated groups for what? (L373-374)
  2. Why the author selected cecal contents but not the other intestinal sections for analysis?
  3. How to identify the cause-and-effect relationship between dysbiosis and lipogenesis? Is there any solid evidence to support the authors' conclusion that antibiotic-induced dysbiosis of gut microbiota promotes the lipogenesis of chicken?
  4. The results shown in Fig. 4 demonstrated that carbohydrate metabolism and lipid metabolism-related microbiota were enriched and amino acid metabolism and nucleotide metabolism-related microbiota were depleted in response to antibiotic treatment. Please provide a section of discussion pertaining to these results.

Author Response

Dear reviewer,

We really appreciate your helpful comments and valuable suggestions. The comments have been addressed and corrected. We hope that the present manuscript now is suitable for publication in Metabolites. All the corrections and modifications are as follows:

Reviewer 1

  1. Five chickens were randomly selected from the control and antibiotic-treated groups for what? (L373-374)

Response: Five chickens were selected from the control and antibiotic-treated groups for sample collection including blood, liver, abdominal fat and cecal contents. For better understanding, this sentence has been moved to the beginning of “4.3. Sample collection” section.

  1. Why the author selected cecal contents but not the other intestinal sections for analysis?

Response: As indicated in the second paragraph, Wen et al. estimated the microbiability of abdominal fat weight and abdominal fat rate of yellow feathered broilers and found that the duodenal and cecal microbiota made greater contributions to fat deposition and could separately account for 24% and 21% of the variance in the abdominal fat mass after correcting for host genetic effects. The above results indicate that cecal microbiota might play an important role in regulating lipogenesis and fat deposition of chicken (Wen, C et al. The gut microbiota is largely independent of host genetics in regulating fat deposition in chickens. ISME J. 2019, 13, 1422-1436.). Huang et al. found that cecum have the highest density of microbiota in chicken (1010-1011/g, see the figure below) (Huang P et al. The chicken gut metagenome and the modulatory effects of plant-derived benzylisoquinoline alkaloids. Microbiome. 2018 Nov 27;6(1):211.). Therefore, we selected cecal contents but not the other intestinal sections for analysis.

  1. How to identify the cause-and-effect relationship between dysbiosis and lipogenesis? Is there any solid evidence to support the authors' conclusion that antibiotic-induced dysbiosis of gut microbiota promotes the lipogenesis of chicken?

Response: Studies have shown that gut microbiota participates in regulating host lipid metabolism and fat storage (Bäckhed F., et al. The gut microbiota as an environmental factor that regulates fat storage. Proc Natl Acad Sci U S A. 2004, 101(44):15718-23. doi: 10.1073/pnas.0407076101). In chicken, the cause-and-effect relationship between gut microbiota and fat deposition was also reported (Wen C, et al. The gut microbiota is largely independent of host genetics in regulating fat deposition in chickens. ISME J. 2019, 13(6):1422-1436./Li H., et al. Propionate inhibits fat deposition via affecting feed intake and modulating gut microbiota in broilers. Poult Sci. 2021,100(1):235-245.). In this study, we found that oral antibiotics caused dysbiosis of cecal microbiota and promoted the lipogenesis of chicken. The relationship between the altered cecal microbiota and lipogenesis were analyzed by Mantel test and Pearson correlation analysis (section 2.7). These results showed significant correlations between cecal microbiota and lipogenesis at phylum and genus levels. However, it is important to note that this cause-and-effect relationship between dysbiosis and lipogenesis in our study is only a primary step and need to be further investigated by other methods such as faecal microbiota transplantation and high-fat diet induction in the future.

  1. The results shown in Fig. 4 demonstrated that carbohydrate metabolism and lipid metabolism-related microbiota were enriched and amino acid metabolism and nucleotide metabolism-related microbiota were depleted in response to antibiotic treatment. Please provide a section of discussion pertaining to these results.

Response: A paragraph of discussion has been added (In this study, Tax4Fun algorithm was used to predict the potential function of the cecal microbiota of chickens. Bacteria involved in carbohydrate metabolism and lipid metabolism-related microbiota were significantly enriched in antibiotic-treated chickens. While bacteria involved in amino acid metabolism and nucleotide metabolism-related microbiota were significantly depleted in response to antibiotic treatment. This suggests that the antibiotic-induced dysbiosis of cecal microbiota might promote carbohydrate metabolism and lipid metabolism and inhibit amino acid metabolism and nucleotide metabolism of chicken.).

Reviewer 2 Report

Antibiotic-induced dysbiosis of microbiota promotes chicken 2 lipogenesis by altering metabolomics in the cecum 3

Tao Zhang presents a deep analysis of the fat metabolism in antibiotic treated birds in comparison to control animals. The work is well presented and contributes to understand the secondary effects of antibiotic treatment in meat animal production.

Minor comments

In figure 2C, one of the labels seems incorrect is NFD-1 NFSA-1?

In figure 7.8 and 9 the for the Correlation network the thickness of the line representing the degree of correlation, this difference is not easy to visualize, the reviewer suggest a different way to visualize the degree of correlation, or alternatively to include the correlations as supplementary material in a tabular form.

Regarding the number of chickens, it is not clear why in this experiment out of 10 experimental chickens only 5 were chosen for some of the studies. The authors should ass the n= number for each method and better explain in which phase of the experiment 10 chickens were used and which ones only used 5 chickens and present a rational for the use of only 5 chickens in  some analysis.

English should be revised, wrong verb conjugation and phrases hard to understand are present in the text for example “The raw data was performed to quality control using FASTP software”; use of faces instead of faeces.

Citation of some software used seems to be incomplete (eg. Tax4Fun was used to predict the function of the cecal microbiota of chickens in control and antibiotic treatment groups) or  MetaboAnalyst 5.0.was used; it is recommended that all software citations be checked and when possible guidelines be followed such as Citing the paper describing the software  or Citing a DOI for the software, for example, obtained via Zenodo or FigShare (e.g. Foreman-Mackey et al. 2014, corner.py, v0.1.1, Zenodo, doi:10.5281/zenodo.11020, as developed on GitHub).

Author Response

Dear reviewer,

We really appreciate your helpful comments and valuable suggestions. The comments have been addressed and corrected. We hope that the present manuscript now is suitable for publication in Metabolites. All the corrections and modifications are as follows:

  1. In figure 2C, one of the labels seems incorrect is NFD-1 NFSA-1?

Response: It should be NFD-1. Figure 2C displays the clustering heatmap of samples based on all metabolic peaks. Theoretically, sample NFD-1 should be clustered together with the other four samples in the control group (NFD-2, NFD-3, NFD-4 and NFD-5). However, the clustering heatmap showed that NFD-1 was clustered together with samples in the antibiotic-treated group, suggesting that NFD-1 was an outlier sample and was excluded from subsequent analysis.

  1. In figure 7.8 and 9 the for the Correlation network the thickness of the line representing the degree of correlation, this difference is not easy to visualize, the reviewer suggest a different way to visualize the degree of correlation, or alternatively to include the correlations as supplementary material in a tabular form.

Response: Thanks for the helpful suggestion. We changed the way to present the degree of correlation by including the correlations as supplementary tables (Table S4, Table S5 and Table S6).

  1. Regarding the number of chickens, it is not clear why in this experiment out of 10 experimental chickens only 5 were chosen for some of the studies. The authors should ass the n= number for each method and better explain in which phase of the experiment 10 chickens were used and which ones only used 5 chickens and present a rational for the use of only 5 chickens in some analysis.

Response: At the beginning, twenty 4-week old chickens with similar body weight were selected and randomly divided into two groups, control (n = 10) and antibiotic-treated (n = 10), respectively. After 21 days of antibiotic treatment, only ten chickens were selected for subsequent analysis. This is because few of the chickens may experience various questions such as disease and stress. To guarantee a sufficient sample size in the subsequent analysis, we chose a relatively large sample size at the beginning. After 21 days of antibiotic treatment, five healthy chickens with similar body weight within the antibiotic-treated group were selected, and five healthy chickens with similar body weight within the control group were selected. A total of ten chickens were subjected to lipogenesis, microbiome and metabolomics analysis: control (n = 5) and antibiotic-treated (n =5). The sample size for each analysis was added to the text and figure caption.

  1. English should be revised, wrong verb conjugation and phrases hard to understand are present in the text for example “The raw data was performed to quality control using FASTP software”; use of faces instead of faeces.

Response: The English language of the present manuscript has been improved by a native English-speaking colleague.

  1. Citation of some software used seems to be incomplete (eg. Tax4Fun was used to predict the function of the cecal microbiota of chickens in control and antibiotic treatment groups) or MetaboAnalyst 5.0.was used; it is recommended that all software citations be checked and when possible guidelines be followed such as Citing the paper describing the software or Citing a DOI for the software, for example, obtained via Zenodo or FigShare (e.g. Foreman-Mackey et al. 2014, corner.py, v0.1.1, Zenodo, doi:10.5281/zenodo.11020, as developed on GitHub).

Response: We have checked all software citations throughout the manuscript. All the software are followed by a reference, doi number or website.
